# Antimicrobial Properties of TiO_2_ Microparticles Coated with Ca- and Cu-Based Composite Layers

**DOI:** 10.3390/ijms23136888

**Published:** 2022-06-21

**Authors:** Razvan Bucuresteanu, Monica Ionita, Viorel Chihaia, Anton Ficai, Roxana-Doina Trusca, Cornelia-Ioana Ilie, Andrei Kuncser, Alina-Maria Holban, Grigore Mihaescu, Gabriela Petcu, Adela Nicolaev, Ruxandra M. Costescu, Mihai Husch, Viorica Parvulescu, Lia-Mara Ditu

**Affiliations:** 1Microbiology Department, Faculty of Biology, University of Bucharest, Intr. Portocalelor 1-3, 060101 Bucharest, Romania; razvan.bucuresteanu@drd.unibuc.ro (R.B.); alina.m.holban@bio.unibuc.ro (A.-M.H.); grigore.mihaescu@bio.unibuc.ro (G.M.); 2Research Institute of the University of Bucharest, Sos. Panduri 90, 050663 Bucharest, Romania; 3Faculty of Chemical Engineering and Biotechnologies, University Politehnica of Bucharest, 1-7 Gh Polizu Street, 011061 Bucharest, Romania; ionita_monica@yahoo.com; 4Institute of Physical Chemistry “Ilie Murgulescu”, Romanian Academy, Splaiul Independentei 202, 060021 Bucharest, Romania; vchihaia@icf.ro (V.C.); gpetcu@icf.ro (G.P.); 5Department of Science and Engineering of Oxide Materials and Nanomaterials, Faculty of Chemical Engineering and Biotechnologies, University Politehnica of Bucharest, 1-7 Gh Polizu Street, 011061 Bucharest, Romania; anton.ficai@upb.ro (A.F.); cornelia_ioana.ilie@upb.ro (C.-I.I.); 6National Centre for Micro and Nanomaterials and National Centre for Food Safety, Faculty of Chemical Engineering and Biotechnologies, University Politehnica of Bucharest, Spl. Indendentei 313, 060042 Bucharest, Romania; truscaroxana@yahoo.com; 7Academy of Romanian Scientists, 3 Ilfov Street, 050045 Bucharest, Romania; 8National Institute of Materials Physics, 405A Atomistilor Street, 077125 Magurele, Romania; andrei.kuncser@infim.ro (A.K.); adela.nicolaev@infim.ro (A.N.); ruxandra.costescu@infim.ro (R.M.C.); 9Faculty of Building Services Engineering, Technical University of Civil Engineering Bucharest, 020396 Bucharest, Romania; mihai.husch@utcb.ro

**Keywords:** antimicrobial coating, Ca-Cu composite layer, TiO_2_ microparticles, cell wall electric charge

## Abstract

The ability of TiO_2_ to generate reactive oxygen species under UV radiation makes it an efficient candidate in antimicrobial studies. In this context, the preparation of TiO_2_ microparticles coated with Ca- and Cu-based composite layers over which Cu(II), Cu(I), and Cu(0) species were identified is presented here. The obtained materials were characterized by a wide range of analytical methods, such as X-ray diffraction, electron microscopy (TEM, SEM), X-ray photoelectron (XPS), and UV-VIS spectroscopy. The antimicrobial efficiency was evaluated using qualitative and quantitative standard methods and standard clinical microbial strains. A significant aspect of this composite is that the antimicrobial properties were evidenced both in the presence and absence of the light, as result of competition between photo and electrical effects. However, the antibacterial effect was similar in darkness and light for all samples. Because no photocatalytic properties were found in the absence of copper, the results sustain the antibacterial effect of the electric field (generated by the electrostatic potential of the composite layer) both under the dark and in light conditions. In this way, the composite layers supported on the TiO_2_ microparticles’ surface can offer continuous antibacterial protection and do not require the presence of a permanent light source for activation. However, the antimicrobial effect in the dark is more significant and is considered to be the result of the electric field effect generated on the composite layer.

## 1. Introduction

The development of effective and cost-efficient methods for fighting healthcare-associated infections (HAIs) is the main challenge of the 21st century [1]. The pandemic phenomenon of antimicrobial resistance (AMR) has caused an exacerbation of the virulence of pathogens and has forced the discovery of new methods to control HAIs [2,3,4]. Recent research has highlighted the importance of microbial biofilms in the emergence and spread of high-virulence pathogens. One of the best measures to prevent HAIs is to block the development of biofilms on medical surfaces [5,6]. Simultaneously with the development of new active methods for disinfection (discovery of new disinfectants, use of automatic disinfection and ozonation equipment, implementation of robots with UV-C), special emphasis has been placed on the discovery and use of antimicrobial coatings for medical surfaces exposed to viral and microbial pathogen loads [3,7].

Until now, antimicrobial-coated medical devices with anti-adherent properties have been widely implemented, in most cases involving nanoparticles (metal, metal–polymer nanocomposites, bimetallic nanoparticles, and polymer nanoparticles) [8,9,10]. The antimicrobial properties of nanoparticles’ ions (such as Ca^2+^, Zn^2+^, Mg^2+^, Fe^2+^, Fe^3+^, Ni^2+^, and Cu^2+^) have been extensively studied because of their potential to generate ROS or create oxidative stress as the main mechanism of the lethal effect [8]. Moreover, nanomaterial based on chitosan and lysozyme-functionalized magnetite have been proven to be very efficient bioactive nanostructured coatings for medical implants, with good anti-biofilm experimental results [10]. Compositions containing biocidal agents, such as 1,2-benzothiazol-3-one, silver, or copper nanoparticles, are applied on the surfaces of medical devices and supplies [11,12]. These coatings have the role of decreasing the bacterial load by blocking bacterial adherence and biofilm development on these surfaces [10,13,14,15].

Photocatalytic surfaces based on TiO_2_ nanoparticles (NPs) have recently been developed and introduced into current practice [15,16]. Under UV light, TiO_2_ generates a photocatalytic effect that produces reactive oxygen species (ROS) responsible for the degradation of various organic compounds. Being an exogenous mechanism, ROS species also act on bacterial targets without interfering with the metabolic mechanisms of bacteria. Therefore, in principle, the antibacterial photocatalytic effect cannot generate a bacterial resistance response [17]. However, some studies have also showed that TiO_2_ NPs exposed to UV radiation induce adverse effects on eukaryotic cells through a similar mechanism that is antimicrobial efficient: they can react with free radicals and signal molecules, and interfere with the biochemical reactions on plasmalemma, inducing human cell damage, genotoxicity, and inflammation, and exacerbating the immune response [18]. For this reason, current biomedical applications are focusing on the development of TiO_2_ coatings and materials with a local bioactive effect to obtain antimicrobial surfaces while limiting the amount of toxic ROS that can be released into the human body, thus avoiding side effects.

The present study aimed to obtain and characterize new composite layers coated on TiO_2_ microparticles with antimicrobial properties for applications on various devices and surfaces, including ceramics, plastic, glass, stone, or concrete, with great potential in limiting the growth and development of pathogens in health-care facilities. The obtained materials contained commercial TiO_2_ microparticles coated with composite layers based on Ca and Cu and can be applied as a thin film on a plastic surface, showing an efficient antimicrobial effect after at least 2 h of exposure to various microorganisms. The obtained results evidenced a more significant antibacterial effect under dark conditions than light.

## 2. Results

### 2.1. Scanning and Transmission Electron Microscopy

TiO_2_ microparticles were modified by coating a surface with Ca and Cu composite layers. The obtained samples were named S0 (commercial TiO_2_), S1 (sample with 10% Ca on TiO_2_), and S2 (sample with 10% Ca and 2% Cu on TiO_2_). The morphology of the obtained samples was evidenced by scanning electron microscopy (SEM) (Figure 1).

The images of the coated TiO_2_ microparticles were evidenced by transmission electronic microscopy (TEM) (Figure 2).

Electron diffraction in a selected area (SAED) demonstrated the crystalline nature of the rutile TiO_2_ microparticles. The elementary distributions of Ca and Cu on the TiO_2_ microparticles were highlighted by the elementary mappings obtained in scanning transmission electron microscopy (STEM) mode (Figure 3).

### 2.2. X-ray Diffraction

The X-ray diffractogram of the S2 sample (Figure 4) confirms the presence of rutile as the dominant crystalline phase. Thus, the most intense diffraction peaks located at 2θ = 27.2, 36.02, 41.2, 44.06, 56.47, and 69.01 corresponded to TiO_2_ as the rutile phase [19] while the peaks located at 2θ = 29.2 and 38.8 suggest the presence of Cu_2_O and CuO [20]. The other three peaks from 2θ = 11.06, 22.54, and 30.9 are attributed to the hydrocalumite structure. Hydrocalumite is a result of Ca and Al interaction on TiO_2_ microparticles under alkaline conditions. Aluminum ions are commonly used as a stabilizer for commercial rutile TiO_2_ and a basic environment is used for most of the composite layer’s steps. In addition to the identified peaks, there are some that can be attributed to the possible phases of the composite layer.

### 2.3. X-ray Photoelectron Spectroscopy

All samples were analyzed using XPS. We aimed to study the surface of the S1 and S2 samples and analyze the copper on the surface. In Figure 5, a comparison of the XPS spectra of the main elements of the S1 and S2 samples is shown. The Cu 2p spectra show a broad peak deconvoluted in two peaks centered at 931.63 and 932.67 eV that can be attributed to Cu(0)and/or Cu(I) and Cu(II). Because of the peak overlap in energy, it is very difficult to distinguish between Cu(0) and Cu(I) just from the Cu 2p region. Additionally, two satellite peaks were observed at 940.03 and 942.35 eV, which confirms the presence of Cu(II) on this surface. The Cu(I)/Cu(II) ratio of almost 1:2 obtained by deconvolution of Cu 2p3/2 proves that on the surface, there is a mixture of Cu(I) and Cu(II). The Ti 2p XPS spectra (Figure 5) are typical for Ti^4+^ and the two peaks at 457.75 and 463.54 eV indicate the rutile form of Ti 2p. The atomic composition of all samples is presented in Table 1.

The XPS spectra for O 1s exhibited three peaks centered at around 529, 531, and 535.4 eV. The first peak was attributed to O^2−^ from titanium and copper oxides (529 eV) [1,2,20,21], the second to calcium oxide (531 eV) [22,23], and the last to water oxygen species from the surface. Decreases in the intensity of all the peaks and a shift towards higher binding energy values were observed for the S2 sample. For the S2 sample, the significant effect on the intensity of the second peak of the O 1s spectrum confirms the decrease in oxygen from a Ca oxide due to Ca–Cu interaction.

### 2.4. Density Functional Theory Calculations

The isosurfaces of the electrostatic potentials, which were estimated using density functional theory calculations, for the pristine and Ca-deposited TiO_2_(001) surfaces, and Cu_2_O(001)//Ca-TiO_2_(001) interface are presented in Figure 6. The phase diagram database provided by the Materials Project [24]. 

It can be seen that the addition of the Ca on the TiO_2_(001) surface, where Ca atoms form strong bonds with the dangling free oxygen atoms from the top of the TiO_2_(001) surface, determines a perturbation in the averaged electrostatic potential on the surface plane, with about −12.2 eV at the position of the calcium atom compared with about −22.4 eV at the position of the titanium located below the calcium atom. Thus, an external potential perpendicular to the TiO_2_(001) surface is revealed. 

The addition of a thin Cu_2_O(001) film on the top of the Ca-TiO_2_(001) surface induces a large modification of the electrostatic potential. The electrostatic potential, in this case, has important components on the surface plane and the resulting electrostatic potential has an important dependence on the position in space. 

### 2.5. UV-Vis Absorption Spectra

The absorption spectra of pure and Ca or Ca-Cu-modified TiO_2_ samples are shown in Figure 6. All the samples exhibit a broad absorption peak between 230 and 370 nm. The peak of the sample modified with copper (Figure 7, sample S2) extends significantly towards the visible region. 

### 2.6. SEM Microscopy of the Bacterial Cell Membrane

The SEM images (Figure 8) show morphological and numerical changes in the Gram-positive and Gram-negative bacterial cells after 30 min of contact with the pigment samples. The images confirm the cell membrane integrity test, showing for the Gram-negative strain *E. coli* ATCC 25922 a difference in the size of the bacterial cell, which was smaller (1.155 µm) in the case of sample 1 (Figure 8(1C)) compared to sample 2 (2.720 µm) (Figure 8(2C)), and also a lower number of cells distributed on the sample 1-covered surface (Figure 8(1C)) after 30 min of contact.

### 2.7. Antimicrobial Efficacy Tests

Qualitative screening of the antimicrobial activity was used to evaluate the efficiency of the three samples in two different incubation conditions by measuring the diameters of the inhibition zone expressed by each tested microbial strain. The results were expressed as the diameter ratio, statistically analyzed, and are shown in Figure 9 and Figure 10. The lower the ratio values, the higher the antimicrobial effect. 

As shown in Figure 9, the recorded values for the diameter ratio demonstrate statistically significant differences for *E. coli* Gram-negative strains when the incubation was performed in darkness, compared with the light condition, in the presence of the S1 sample. Moreover, the S1 sample demonstrated significantly lower ratio values than the 0 sample and more bass than the S2 sample. These results suggest that photocatalytic activation of the substrate cannot be considered as the only antimicrobial mechanism.

For *P. aeruginosa* 494 strains, the difference in the diameter ratio between the two incubation conditions is not significant, with both the S1 and S2 samples expressing an equal inhibition zone. Compared with the 0 sample (TiO_2_ rutile), both samples demonstrated significantly lower ratio values of the inhibition zone diameters.

For the tested *S. aureus* strains, there was a significant difference in the value of the inhibition diameters in the two incubation conditions, with sample 1 inducing a greater inhibitory effect after incubation in the light compared to incubation in darkness (Figure 10) and compared to the S2 sample. For the other Gram-positive strains (*C. albicans* and *E. faecalis*), the results were variable, demonstrating that the antimicrobial efficiency depends on the cell wall structure and the molecular membrane electric charge, in direct relation to the chemical composition of the pigment substrate. 

The antimicrobial activity of the tested samples was quantitatively evaluated by covering a surface with a uniform layer of each sample over which the bacterial suspensions were added, followed by determination of the viable cell number after 2 h of contact. The results were statistically processed and integrated into Figure 10. The viable cell count (VCC) method was performed (according to the standard method ISO 22196/2011) using two test conditions (visible light and darkness).

In our experiment, the tested sample 1 (10% Ca^2+^-decorated TiO_2_ rutile) and the S2 sample (10% Ca^2+^-decorated TiO_2_ rutile + 2% Cu^2+^) significantly inhibited the multiplication of all tested microbial strains, under both incubation conditions (**** *p* < 0.0001) (Figure 11). Both samples showed an inhibitory effect even in the absence of light radiation, which suggests the possibility that the activation of the substrate may not be caused by a photocatalytic effect. 

### 2.8. Membrane Permeability Test

In order to elucidate the possible mechanism of action manifested by the samples on the bacterial cells with which they came into contact, a membrane permeability test was performed. The test is based on the detection of the relative fluorescence intensity of two dyes, N-phenyl-1-naphthylamine (NPN) and propidium iodine (PI), with specific fluorescent emissions.

NPN (N-phenyl-1-naphthylamine) is a hydrophobic dye that fluoresces weakly in aqueous environments and strongly in hydrophobic environments. The increase in the fluorescence intensity of NPN is in direct relation to NPN binding with nonpolar substances after permeabilization of the outer membrane of Gram-negative strains. Analyzing the graphic representation of the fluorescence intensity of NPN, after 30 min of contact of *E. coli* strains (standard and clinical strains) with the samples, it was observed that only *sample 1 (10% Ca^2+^-decorated TiO_2_ rutile)* generated a significant increase in the fluorescence intensity of NPN compared to the control (bacterial suspension not exposed to contact with pigment) and compared to the other two samples (Figure 12). Thus, the change in the fluorescence intensity of NPN reflects the efficacy of sample 1 at increasing the permeability of the Gram-negative bacteria’s outer membrane.

As a response to direct changes in the inner membrane permeability of the Gram-positive strains after contact with the pigment surfaces, detection of the relative fluorescence intensity of PI was performed. PI is a red-fluorescent nucleic acid stain that can bind to DNA and RNA between the bases only if the inner cell membrane is damaged. Binding to DNA and RNA leads to a strong increment in their fluorescence [25].

Considering this, our results expressed in Figure 13 suggest that the tested pigments have a weak influence on the Gram-positive bacterial cell wall structure, with the same sample 1 represented by *10% Ca^2+^-decorated TiO_2_ rutile* generating a significant increase in the fluorescence intensity of PI only for one of the two tested *S. aureus* strains.

## 3. Discussion

Research studies have recently focused on the development of composite materials based on TiO_2_ as powders or layers with photocatalytic antibacterial activity in order to eliminate the disadvantages of the frequently used biotoxic agents [15,16,26,27]. Since the discovery of the photocatalytic effect in 1970 by Akira Fujishimaet al. [28], it has been observed that photocatalytically activated TiO_2_ triggers a series of chemical reactions that result in reactive oxygen species (ROSs) with a biotoxic role [15]. Various attempts have been made to construct photocatalytic compositions, including ceramic pieces [7]. TiO_2_ as a photocatalytic material has the disadvantage of being activated only under UV light, which is toxic to the human body. In the absence of radiation, or if an obstacle is placed between the surface covered with catalytic material and the light source, the antibacterial effect usually ceases, and the surface may be repopulated with pathogens.

New composite layers with Ca and Cu were obtained by coating TiO_2_ surface microparticles. The aim of this study was to obtain an efficient antimicrobial agent that can eliminate the disadvantages of the current similar materials in terms of the photocatalytic effect-dependent antimicrobial activity [17,29].

The SEM images showed (Figure 1) a disordered spherical morphology for all the samples. This is the morphology of commercial TiO_2_ microparticles with sizes between 150 and 300 nm. The TEM images (Figure 2) confirmed the variation in the microparticles’ size and morphology. Between the spherical and elliptical microparticles of the coated TiO_2_, the presence of another phase was also evidenced. This indicates that in addition to the composite layer from TiO_2_ microparticles, there may also be traces of unsupported composite. The XRD patterns evidenced the presence of a rutile phase for all the samples (Figure 4). In order to check the chemical composition on the surface and Cu oxidation states, samples were investigated using XPS. The obtained XPS spectrum (Figure 4) confirmed the presence of Ti^4+^ species and oxygen as O^2−^ in titanium, copper, and calcium oxides and adsorbed species from the surface (as water). The presence of Ca(II), Cu(II), Cu(I), and Cu(0) species was also evidenced on the surface of the S2 sample. The presence of CuO, CaO, and other crystalline species was evidenced by XRD (Figure 4).

In the first step, the TiO_2_ crystalloid was coated with calcium ions. The strong interaction of Ca^2+^ and negative sites from the TiO_2_ surface, generated by the basic pH, has a significant influence on the composite layer’s stability and properties. Therefore, for the S1 sample, the increase in the Ca-O bond concentration was highlighted on the surface, but no effect was observed for the Ti^4+^ species (Figure 5). Thus, a pseudo-perovskite-like structure of calcium titanium-perovskite formed on the surface of the TiO_2_ crystalloid. In the second step, Cu^2+^ ions were supported on the first layer of the surface to obtain a composite layer with a higher absorption intensity (Figure 6) and photocatalytic activity in the visible spectral range (400–700 nm). For the Cu-doped TiO_2_ samples, the shift in the absorption above 400 nm was considered as a consequence of the charge transfer from the O 2p to the d states with Cu, and the visible absorption is due to the electronic transition d-d [30]. High absorbance values over a wide visible range of 400 to 800 nm may result from the interaction between the copper ions and surrounding species (e.g., OH^−^, H_2_O). These species are removed by calcinations, and the UV-VIS spectrum of the sample, calcined in air at 800 °C, showed a significant decrease in the intensity of the visible absorption band (Figure 7, S2c sample). The obtained material with copper showed antimicrobial photocatalytic activity, as highlighted in previous publications [17].

The performed DFT calculations show that the adsorbed calcium might result in an electric field perpendicular on the surface and that the addition of a thin film of Cu_2_O on top of the Ca-TiO_2_ system results in a very complex distribution of the electrostatic potentials, with an electric field with components also in the plane of the surface.

The antimicrobial activity of the three samples under the two different incubation conditions demonstrated statistically significant results. Our study showed that the tested S1 sample (10% Ca^2+^-decorated TiO_2_ rutile) and sample 2 (10% Ca^2+^-decorated TiO_2_ rutile + 2% Cu^2+^) significantly inhibited the multiplication of all tested microbial strains under both incubation conditions (Figure 9). Both samples showed an inhibitory effect even in the absence of light radiation, which suggests the possibility that photocatalytic activation of the substrate is not the only antimicrobial mechanism.

The biological activity of the TiO_2_ composite decorated with 10% Ca^2+^ and 2% Cu^2+^, as demonstrated by antimicrobial tests and membrane permeability tests, can be explained based on the interactions between the negative electric charge of the microbial cell wall and the positively charged composite layers supported on the TiO_2_ microparticles.

Gram-positive and Gram-negative bacteria have differences in their membrane structure, with Gram-negative cell walls having an additional outer membrane covered in negatively charged lipopolysaccharides. At the same time, the Gram-positive walls possess only the inner membrane, with a representative peptidoglycan layer containing negatively charged teichoic acids [31,32]. In conclusion, most microorganisms have a negative charge when cultivated at physiological pH values [31].

The SEM images highlight these electrostatic interactions due to the presence of bacterial cells adhering to the surface of the composites 30 min after contact (Figure 8). However, the electrostatic attraction of these two surfaces is not enough to generate the destruction of the cell wall. The large modification of the S1 sample’s surface electrostatic potential may induce cell membrane potential changes, which explains the permeability of the outer membrane of the *E coli* strain as demonstrated by the increase in the fluorescent signal for NPN (Figure 12).

On the other hand, the S1 sample (*10% Ca^2+^-decorated TiO_2_ rutile*) generated a significant increase in the fluorescence intensity of PI (propidium iodine) only for the standard *S. aureus* ATCC 25923 strain but not for the clinical strain *S. aureus* MRSA 5578. If, for the standard strain, the permeabilization effect of the cell wall and the inner membrane can be explained by the same mechanism described above, for the MRSA clinical strain, the lack of a fluorescent signal for PI and, implicitly, the lack of a permeabilization effect can be explained by the ability of this clinical strain to influence the bacterial microenvironment through the cell wall surface appendages and proteins and cellular secretions [33,34].

As can be seen in the UV-Vis spectrum (Figure 7), only the sample containing Ca and Cu showed absorption in visible light. The antimicrobial photocatalytic effect of copper was highlighted for similar materials [17]. However, this study evidenced a similar antibacterial effect under dark and light conditions for all the samples. Given that only the sample with Ca and Cu generated a photocatalytic effect and the results were similar (Figure 11) we concluded that although it competition may occur between the two processes, the one that determines the antimicrobial effect is that of the electric field (generated by the electrostatic potential of the composite layer, see Section 2.4), which is present both in the dark and in the light.

## 4. Materials and Methods

### 4.1. Synthesis of Composite Layers with Ca and Cu on TiO_2_ Microparticles

Tytanopol TiO_2_ microparticles and Sigma-Aldrich Ca(OH)_2_, CuSO_4_, and NaOH were used as starting materials for the preparation of supported composite layers. A two-step wet method was used for this, described in detail in the patent application PCT/RO2022/05005 [34]. In the first step, 18.5 g of Ca(OH)_2_ was mixed with 500 mL of 1 M NaOH aqueous solution under constant stirring. After 15 min, 100 g of rutile TiO_2_ microparticles was dispersed into the obtained solution. The milky-white suspension was then heated, under continuous stirring, to boiling for 1 h, cooled, and decanted. The obtained sample was named S1. In the second step, a solution containing copper was obtained out of two others containing 8 g of CuSO_4_, dissolved in 150 mL of distilled water, and 2.2 g of NaOH added to 150 mL of H_2_O. Then, 10 mL from sample S1 was dispersed into the obtained solution, heated under stirring for 1 h, cooled, and decanted. The obtained sample was named sample S2.

### 4.2. Scanning Electron Microscopy

The electron microscopy images were obtained using a Quanta Inspect F50 (FEI Company, Eindhoven, The Netherlands) equipped with a field emission gun (FEG) with a 1.2 nm resolution and an energy-dispersive X-ray spectrometer (EDS) with an MnK resolution of 133 eV Kα.

### 4.3. Transmission Electron Microscopy

Transmission electron microscopy (TEM) investigations were performed using a JEOL 2100 instrument (LaB6 electron gun) equipped with a high-resolution polar piece and a JEOL Energy Dispersive X-ray Spectrometer (EDS).

### 4.4. X-ray Diffraction

The composition and crystallinity of the Ca-Cu-modified TiO_2_ powder were analyzed using wide-angle X-ray diffraction using of a Rigaku Ultima IV diffractometer with Cu Kα (λ = 0.15406 nm).

### 4.5. X-ray Photoelectron Spectroscopy

X-ray photoelectron spectroscopy (XPS) measurements were performed in an AXIS Ultra DLD (Kratos Surface Analysis) setup, using Al K_α1_ (1486.74 eV) radiation produced by a monochromatized X-ray source at an operating power of 240 W (12 kV × 12 mA). The base pressure in the analysis chamber was 1.0 × 10^–8^ mbar. Charge compensation was reached with the help of a flood gun, which operated at a 1.7 A filament current, 2.8 V charge balance, and 2.0 V filament bias. High-resolution core-level spectra for all elements were acquired using the hybrid lens mode, 50 eV pass energy, and slot aperture. In the deconvolution of the core-level spectra, Voigt profiles, singlets, or doublets were used, based on the methods described in [3,35] The binding energy scale for all spectra was calibrated to the C1s standard value of 284.6 eV. The atomic composition was determined using the integral areas provided by the deconvolution procedure normalized to the atomic sensitivity factors given by [4,36] The surface contamination with C1s was not considered for the atomic composition.

### 4.6. Density Functional Theory Calculations

The DFT calculation scheme has an important effect on the accuracy of the results [37] For the present study, we compared the accuracy and the computational effort. We performed the spin-polarized DFT calculations using the solid-state code CASTEP [38] using the exchange-correlation potential PBE, the on-the-fly generation ultrasoft pseudopotential OTFG [39] an energy cut-off of 490 eV, and a fine grid for the reciprocal space with a separation of at least 0.04 Å^−1^. The thresholds of the parameters that control the electronic structure included an SCF tolerance of 10^−5^ eV/atom, an electronic minimizer of the type All Bands/EDFT, and a convergence window of 3. The total spin was optimized after each of the 5 SCF iterations. For good treatment of the van der Waals interaction between the different layers of materials, the dispersion corrections TS was introduced with the default DFT-D parameters included in the CASTEP code. The chosen DFT calculation scheme was accurate as the optimized unit cells of the CuO, CuO_2_, Cu_2_O, and rutile-TiO_2_ had very close parameters to the corresponding experimental data.

### 4.7. UV-Vis Spectroscopy

The UV-Vis absorption spectra of the modified TiO_2_ samples were obtained in diffuse reflectance (DRS) mode using a JASCO V570 spectrophotometer.

### 4.8. Antimicrobial Efficacy Tests

The antimicrobial pigment efficacy test was performed according to qualitative and quantitative standards methods with specific adapted steps, using both clinical and standard microbial strains that were included in the microbial collection of the University of Bucharest, Faculty of Biology, Microbiology Department: *Staphylococcus aureus* ATCC 25923 and *Staphylococcus aureus* MRSA 5579 (clinical isolate), *Escherichia coli* ATCC 25922 and *Escherichia coli* ESBL 135 (clinical isolate), *Pseudomonas aeruginosa* ATCC 27853 and *Pseudomonas aeruginosa* 494 (clinical isolate), *Candida albicans* ATCC 10231, and *Candida albicans* 6853 (clinical isolate). All strains were grown on a nutritious agar medium using incubation for 18–20 h at 37 °C to obtain fresh cultures.

For the qualitative screening of the antimicrobial efficiency, an adapted spot diffusion method was used, according to the CLSI standard (Clinical Laboratory Standard Institute, 2021) [40] Microbial suspensions of 1.5 × 10^8^ CFU/ml density (corresponding with 0.5 McFarland standard) were prepared from all microbial cultures and seeded on the agar medium (Müller-Hinton agar for bacterial strains and Sabouraud agar for yeast strain), followed by spotting 50 µL of each sample over the inoculated medium: sample 0 (rutile TiO_2_); sample 1 (10% Ca^2+^-decorated TiO_2_ rutile); and sample 2 (10% Ca^2+^-decorated TiO_2_ rutile + 2% Cu^2+^).

After equal diffusion of the compound in the medium, at room temperature, the plates were incubated at 37 °C for 2 h, in 2 different conditions: visible light and darkness. For the qualitative evaluation of the anti-microbial efficiency, the diameters of the inhibition zones were measured. The results were expressed as the ratio of the droplet diameter to the total diameter (droplet diameter + area of inhibition) to avoid variation between the sample droplet diameters. The lower the ratio, the higher the antimicrobial effect.

The quantitative evaluation of the antimicrobial efficiency was performed based on the ISO 22196/2011 standard method [17,41] In total, 1 mL of each sample was spread on the 6-well plates, maintained for 24 h for drying, and inoculated with 500 µL of the microbial suspensions prepared in 1/500 diluted nutrient broth (distillate water/nutrient broth = 1/500), with a final density of 1.5 × 10^6^ CFU/mL. The inoculated 6-well plates were divided into 3 sets. Set 1 (considered T2h light) was incubated at 37 °C for 2 h in visible light and set 2 (considered T2h darkness) was incubated at 37 °C for 2 h in darkness. A third set (set 3) of 6-well plates covered with the tested samples was inoculated with 500 µL of the microbial suspension and used as the microbial cell recovery control, with no incubation time (considered T0h). In the following steps, all tested sets were treated similarly by adding 1 mL of neutralizing agent (1% sodium thiosulphate), followed by viable cell count determination using a 10-fold serial microdilution method.

### 4.9. Membrane Permeability Test

The integrity of the inner and outer cell membranes of the Gram-positive and Gram-negative bacterial strains was evaluated using an adapted NPN/PI(N-phenyl-1-naphthylamine/propidium iodide) assay that has been described previously by Ma et al. [42]. The tested bacterial strains were represented by: *Staphylococcus aureus* ATCC 25923 and *Staphylococcus aureus* MRSA 5579 clinical isolate; *Escherichia coli* ATCC 25922; and *Escherichia coli* ESBL 135 clinical isolate.

Gram-positive and Gram-negative bacterial cells in the exponential growth phase were suspended in LB broth media (Luria Bertani), at a final density corresponding to 1–1.5 × 10^8^ cells/mL. Subsequently, 2 mL of this suspensions was incubated in the presence of the three pigment samples (rutile TiO_2_; 10% Ca^2+^-decorated TiO_2_ rutile; 10% Ca^2+^-decorated TiO_2_ rutile + 2% Cu^2+^) at 37 °C for 30 min under darkness conditions. After the incubation, the recovered cell suspensions were washed two times with phosphate-buffered saline (PBS, pH 7.4), harvested by centrifugation (11,000× *g* rpm) for 10 min, and resuspended in 2 mL of PBS-NPN-PI solution (pH 7.4 PBS, 20 µM/mL NPN and 5 µg/mL PI). After 10 min of incubation at room temperature and in darkness, cell suspensions were analyzed using a fluorescent spectrophotometer SYNERGY HTX (multi-mode reader). Fluorescence was detected with an excitation of 350 nm and an emission of 450 nm (for NPN), respectively, and an excitation of 535 nm and an emission of 620 nm (for PI).

### 4.10. Statistical Analysis

The antimicrobial test results were statistically analyzed using the GraphPad Prism 9 program, developed by GraphPad Software, San Diego, CA, USA. All experiments were performed in three independent determinations.

The membrane permeability test results were statistically analyzed using a two-way analysis of variance (two-way ANOVA) and Dunnett’s multiple comparisons test. Significance differences were noted as: * for *p* < 0.05, ** for *p* < 0.01, ****p* < 0.001, or **** for *p* < 0.0001 and were considered statistically significant.

## 5. Conclusions

This study reports the synthesis of new composite layers with Ca- and Cu-coated TiO_2_ microparticles. The obtained material with copper showed a broad absorption band with high intensity in visible light. A significant aspect of this composite is that similar antimicrobial effects were evidenced both in the presence and absence of the light as a result of competition between the photo and electrical degradation processes. Because no photocatalytic properties were found in the absence of copper, the results support the antibacterial effect of the layer with Ca due to as effect of the electric field (generated by the electrostatic potential of the composite layer, see Section 2.4) under both the dark and in light conditions. In this way, the composite layers can offer permanent antibacterial protection and do not require the existence of a permanent light source for activation. The obtained aqueous mixture could be applied on various abiotic surfaces in health-care facilities and represents a versatile solution for limiting nosocomial infections.

## Figures and Tables

**Figure 1 ijms-23-06888-f001:**
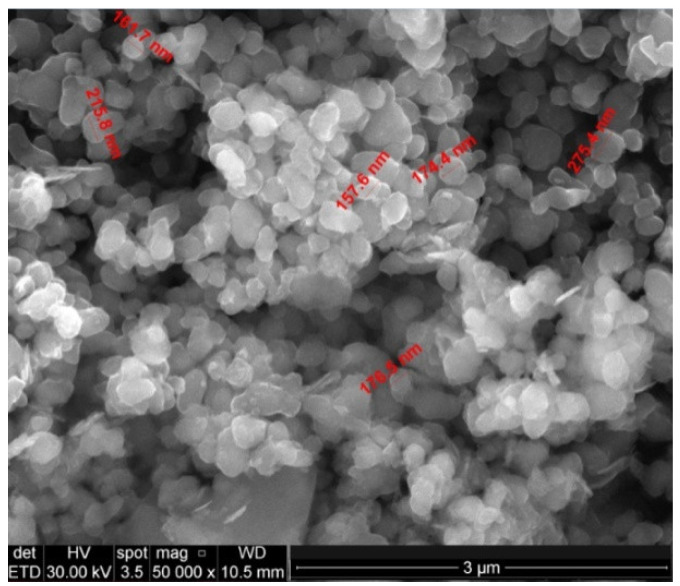
SEM image of the S2 (Ca-Cu/TiO_2_) sample.

**Figure 2 ijms-23-06888-f002:**
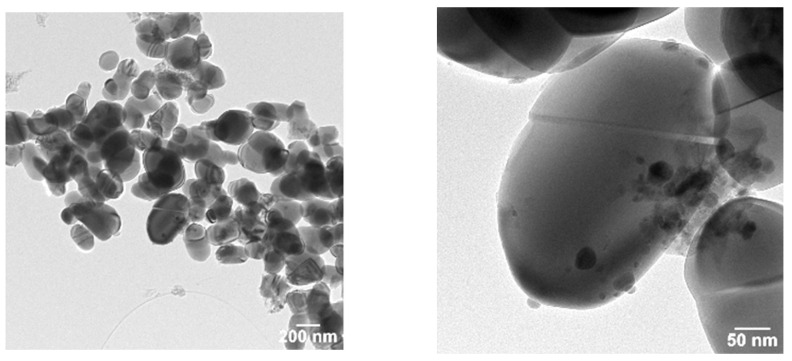
TEM images of the S2 (Ca-Cu/TiO_2_) sample.

**Figure 3 ijms-23-06888-f003:**
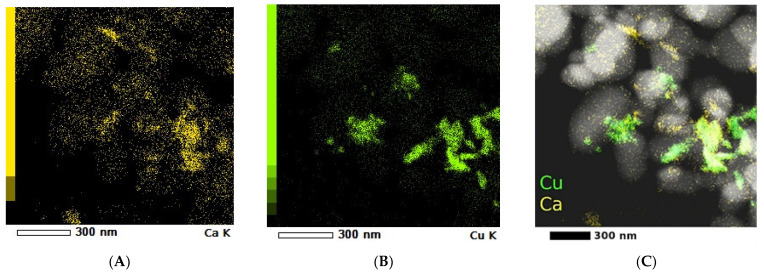
STEM dark-field images for sample S2: (**A**) distribution of Ca; (**B**) distribution of Cu; (**C**) distribution of Ca and Cu.

**Figure 4 ijms-23-06888-f004:**
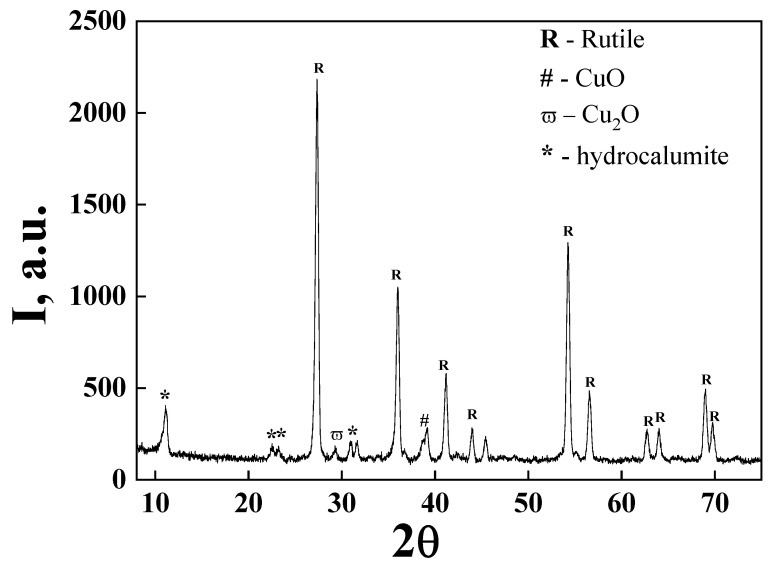
X-ray diffractogram of the S2 (Ca-Cu/TiO_2_) sample.

**Figure 5 ijms-23-06888-f005:**
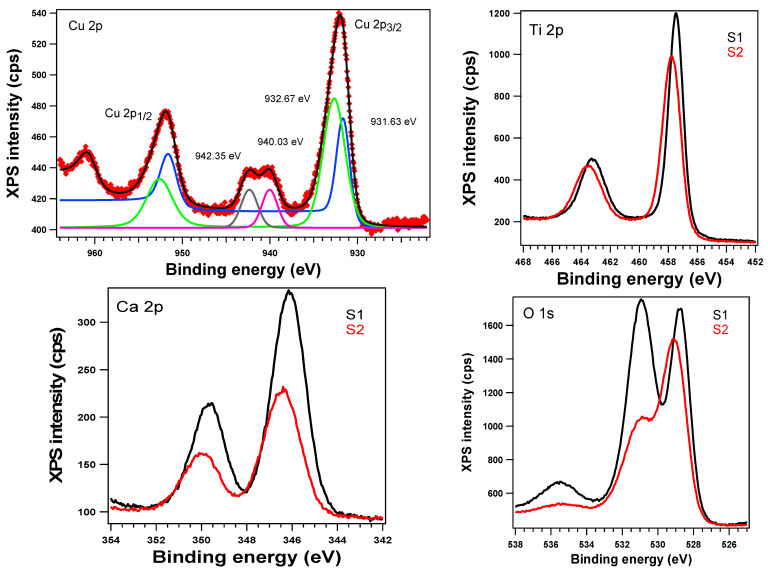
XPS spectra of Cu 2p_3/2_, Ti 2p, Ca 2p, and O 1s for the samples S1 and S2.

**Figure 6 ijms-23-06888-f006:**
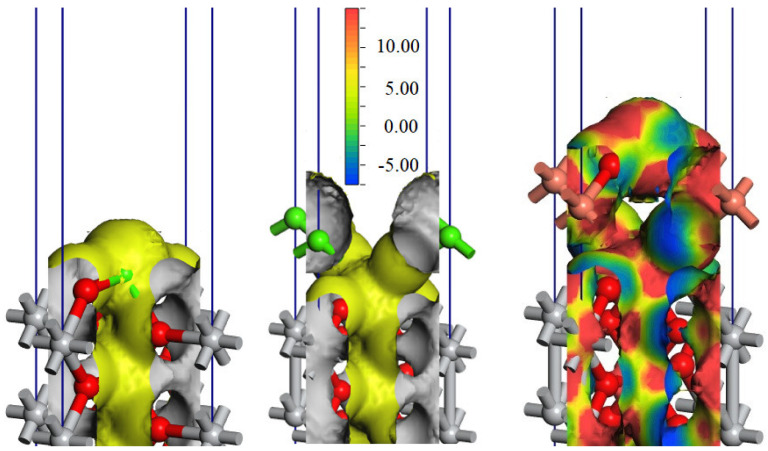
The isosurfaces of the electrostatic potential for pristine TiO_2_(001) (**left**), Ca-deposited TiO_2_(001) surfaces (**center**), and the thin film of Cu_2_O(001) formed on Ca-TiO_2_(001) (**right**), where TiO_2_ is considered with the structure of rutile polymorph. The color map is given for values between −7.5 (blue) and 15 eV (red), with the yellow color corresponding to 5 eV.

**Figure 7 ijms-23-06888-f007:**
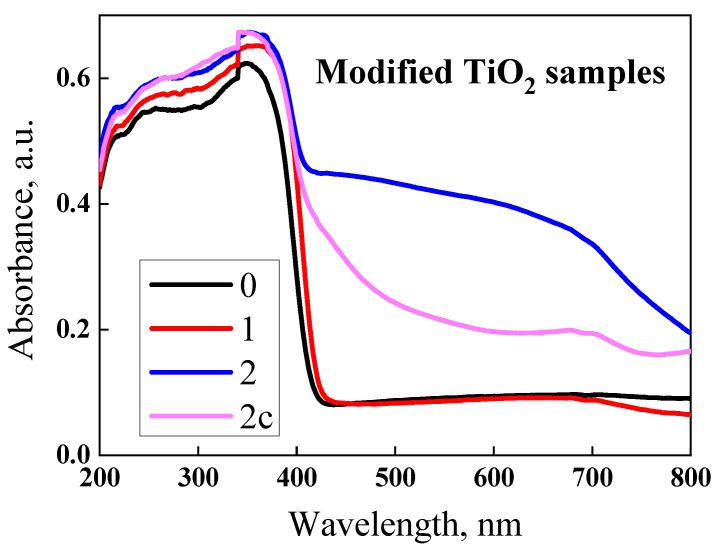
Comparative UV-VIS spectra for the samples: TiO_2_ rutile (S0), TiO_2_ decorated with 10% Ca^2+^ (S1), and TiO_2_ decorated with 10% Ca^2+^ and 2% Cu^2+^ uncalcined (S2) and calcined (S2c).

**Figure 8 ijms-23-06888-f008:**
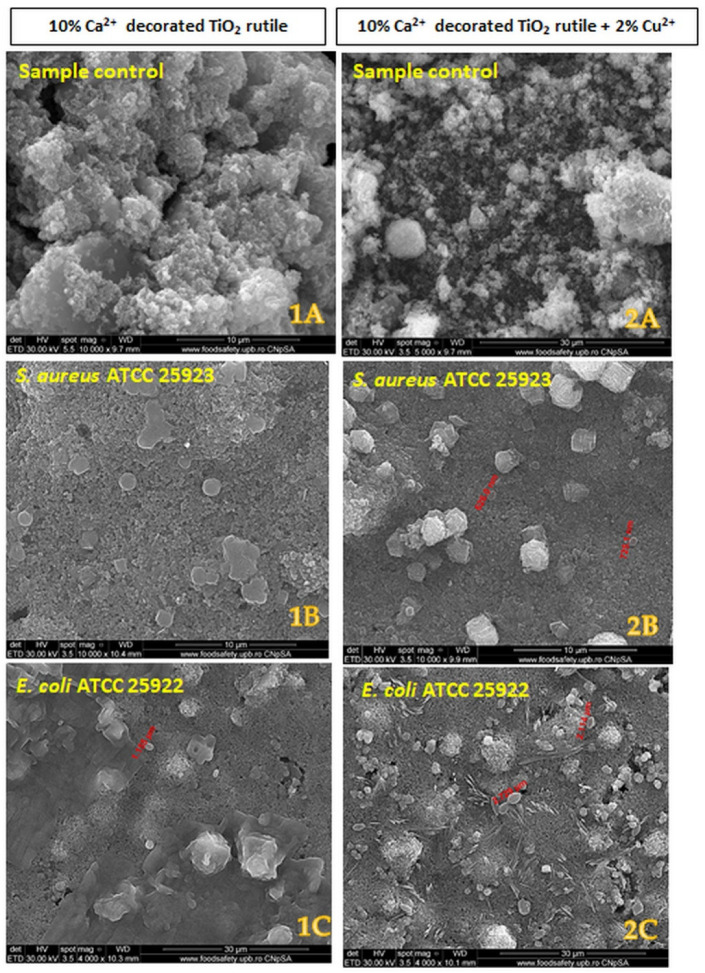
Scanning electron micrograph of the two pigments before and after 30 min of contact with the Gram-positive and Gram-negative bacterial cells. The **left column** represents sample 1; the **right column** represents sample 2. The bacterial strains are listed on the images. The diameters of the bacterial cells are marked on each image: (**1A**) and (**2A**)—controls (samples 1 and 2 that aren’t put in contact with bacterial cells); (**1B**)—no bacterial cells observed; (**2B**)—626 nm; (**1C**)—1.155 µm; (**2C**)—2.720 and 2.114 µm.

**Figure 9 ijms-23-06888-f009:**
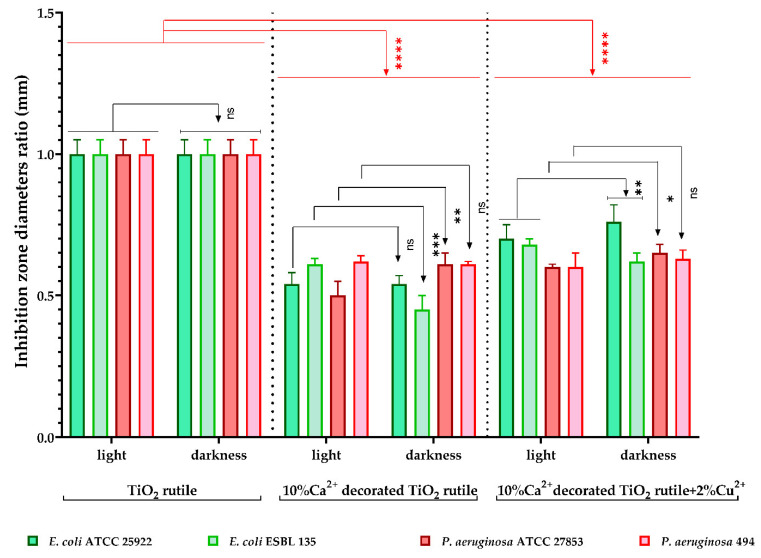
Graphic representation of the inhibition zone diameters, expressed as the diameter ratio, obtained for Gram-negative tested strains after incubation in two different conditions: visible light and darkness. The results were compared using two-way ANOVA and Dunnett’s multiple comparisons tests; ns—not significant; * *p* < 0.05; ** *p* < 0.009; *** *p* = 0.0009; **** *p* < 0.0001.

**Figure 10 ijms-23-06888-f010:**
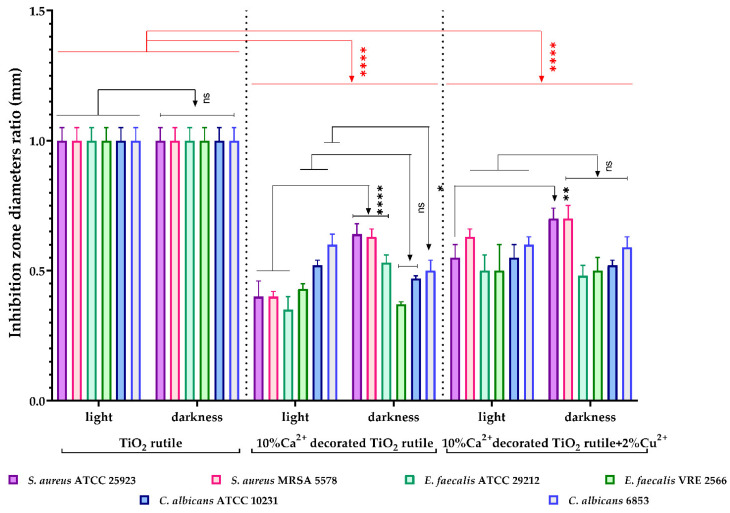
Graphic representation of the inhibition zone diameters expressed as the diameter ratio, obtained for the tested *C. albicans* and Gram-positive strains, after incubation in two different conditions: visible light and darkness. The results were compared using two-way ANOVA and Dunnett’s multiple comparisons tests; ns—not significant; * *p* < 0.05; ** *p* < 0.003; **** *p* < 0.0001.

**Figure 11 ijms-23-06888-f011:**
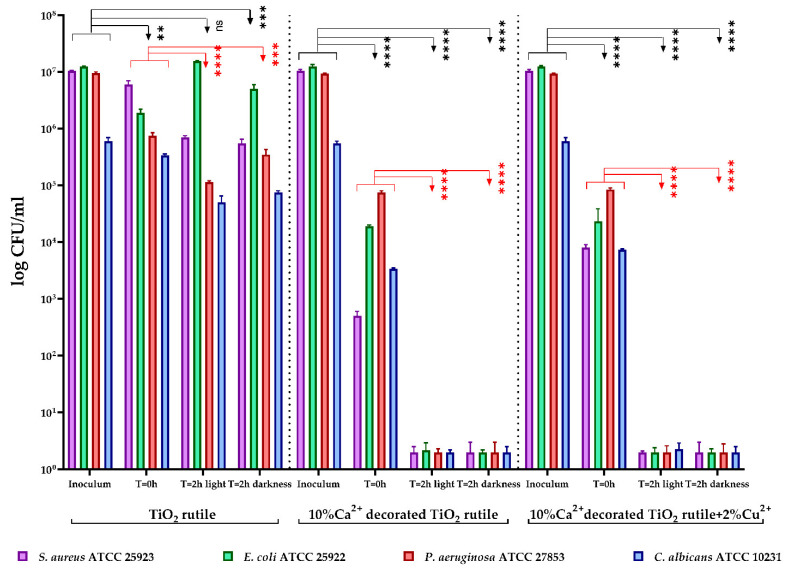
Graphical representation of the log10 values of colony-forming units (CFU)/mL representing the viable cell number after 2 h of contact with the tested products under two conditions: visible light and darkness. The results were compared using two-way ANOVA and Dunnett’s multiple comparisons tests. The results were considered statistically significant (ns—not significant, ** *p* = 0.0011; *** *p* < 0.0004; **** *p* < 0.0001).

**Figure 12 ijms-23-06888-f012:**
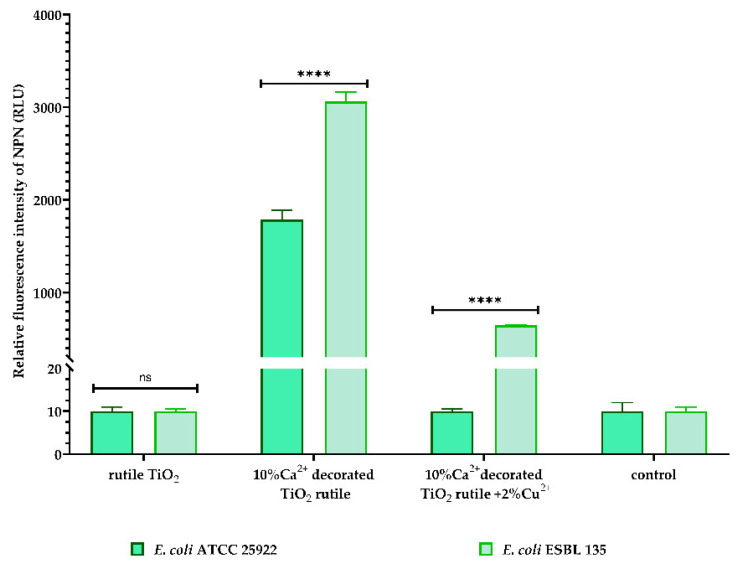
Outer membrane permeabilization of *E coli* strains measured by detecting the fluorescence intensity of NPN after 30 min of contact with the samples. All the data are shown as the mean ± SD of the three independent experiments. *p*-values were determined via two-way ANOVA and Dunnett’s multiple comparisons tests (ns—not significant, **** *p* < 0.0001 vs. untreated control).

**Figure 13 ijms-23-06888-f013:**
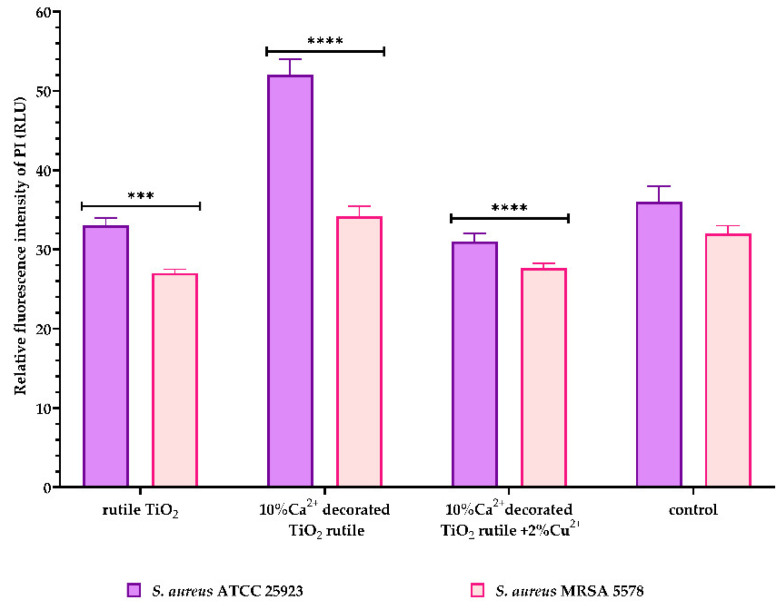
*S. aureus* strains’ inner membrane permeabilization measured by the detection of the fluorescence intensity of PI. All the data are shown as the mean ± SD of three independent experiments. *p*-values were determined via two-way ANOVA and Dunnett’s multiple comparisons tests (*** *p* = 0.003; **** *p*< 0.0001 vs. untreated control).

**Table 1 ijms-23-06888-t001:** Atomic composition (%) of the surface of each sample.

**Chemical Element**	**S0**	**S1**	**S2**
O 1s	88.21	81.7	70.1
Ti 2p	11.79	13.49	16.12
Ca 2p	-	4.8	3.77
Cu 2p	-	-	1.91

## Data Availability

Not applicable.

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
