# Peer review of "Antimicrobial Properties of TiO2 Microparticles Coated with Ca- and Cu-Based Composite Layers"

_ijms, 2022, doi:10.3390/ijms23136888_

Round 1

Reviewer 1 Report

I have included some minor remarks in the attachment.

Author Response

Dear Editor and Reviewers,

We would like to express our sincere gratitude to you and the reviewers for the effort, and precious time you spent reviewing our paper and providing valuable comments in your reports. It was your valuable and insightful comments that helped us to improve our review substantially. The authors have carefully considered the comments and tried their best to fulfil every one of them. We honestly hope that the manuscript meets your high standards after careful revisions. We have also included a point-by-point response to the reviewers and made the changes described above in the manuscript.

Response to Reviewer 1

I highly estimate the novelty, performed research, a deep discussion at a reasonable scientific level, and possible application of developed coating. As concerns the contents of the manuscript and  its quality, I have only some minor remarks:

  • Line 62: ‘involving nanoparticles [8-10]”. Please develop this phrase indicating the kinds of NPs.
  • Response: Thank you very much for your observation. We introduced new details regarding the type of the NPs: Until now, antimicrobial coated medical devices with anti-adherent properties have been widely implemented, in most cases involving nanoparticles (metal, metal-polymer nanocomposites, bimetallic nanoparticles and polymer nanoparticles)[8-10]. It has been extensively studied the antimicrobial properties of the metal nanoparticles ions (such as Ca2+, Zn2+, Mg2+, Fe2+, Fe3+, Ni2+, and Cu2+) because of their potential to generate ROS or to create oxidative stress as main mechanism of the lethal effect [8]. Also, nanomaterial based on chitosan and lysozyme functionalized magnetite proved to be very efficient bioactive nanostructured coatings for medical implants, with good anti-biofilm experimental results [10].”

  • Lines 63 and 243: I do not think that names of chemicals should start with capital letters (Silver, Copper, Benzothiazol, Propidium).
  • Response: We modified the capital letters with lower case.

  • Line 317-318: “The use of rutile microparticles was preferred in order of their future applications as paint pigment”. I understand not all relation between this study and the possible application of rutile microparticles and suggest deleting this sentence.
  • Response: Thank you for your suggestions. We deleted the sentence.

  • Lines 318-321: The authors demonstrate the difference between NPs of rutile and anatase as regards the photocatalytic effect. However, it is still true for microparticles? Whether such effects are observed for such great particles? I am not sure and the authors have used only micronized titania.
  • Response: Tytanopol TiO2 microparticles and Sigma-Aldrich Ca(OH)2, CuSO4 and NaOH were used as starting materials for the preparation of supported composite layers.

  • Line 327: “traces of free composite”. What does it mean, a free composite, and what composite?
  • Response: Thank you very much for your observation. We replaced „traces of free composite” with „traces of unsupported composite”

  • “Materials and methods” section needs a careful examination, as concerns the precision and a language, in particular, subchapters 4.1, 4.7, 4.8.
  • Response: Thank you for your constructive observation. We carefully modified the “Materials and methods” section, by improving the language and clarified some explanation regarding the work methodology.

  • Grammar and linguistic errors
  • Response: Thank you for your observations. We corrected all the grammar and linguistic errors in the suggested lines, except Line 51/Line 52: “healthcare-associated”; rather “health-care related” Healthcare-associated infection is a generally accepted term for nosocomial infections and more frequently used than “health-care related” infection. We prefer to keep this term.

Reviewer 2 Report

Dear authors,

the paper provides interesting results on antibacterial properties of the core-shell systems but there are some inconsistencies in interpretation of diffraction and XPS data.

So, you have found chlorine containing phases (hydrophilite and hydrocalumite on Fig. 4) but chlorine on the surface of particles was not found by XPS or its spectrum was not presented in the paper.  So this point should be discussed in the new version of the paper. In particular, you shall check once more correctness of phases identification by XRD: every phase shall be identified basing on all characteristic lines not only one, these lines should be also marked even when they overlap lines of other phases.

It is necessary to provide the survey XPS spectrum as a figure in the paper before spectra of individual lines and upload this spectrum file and files of individual lines as a supplementary materials if possible in accordance to the good electron spectroscopy traditions.

In general, the whole section presenting XPS results should be significantly improved. Firstly I shall mention that selected spectrometer analyser mode with CAE = 50 eV cannot provide real high energy resolution: high resolution spectra could be recorded at CAE 5 to 10 but such mode requires much more spectra acquisition time to get good signal/noise ratio. From other hand, the selected resolution was adequate for this research, so please remove the words "high resolution" from the text.

On my mind, all figures from Appendix I should be moved to main text where they are discussed (if it does not exceed paper limits). All these figures should be presented in bigger size in the new version of the paper. Components of peaks corresponding to definite interatomic bonds should be labelled on spectra figures.

It is noticed in line 138  "two satellite peaks were observed at 438.53 eV and
439.23 eV" so it is necessary to show them on the figure and discuss its origin from Cu2+ in greater detail: according to NIST XPS database these lines could be Ca 2s.

The paper contains too many inaccuracies, style errors and misprints.

In the beginning of Section 2 "Results" the samples are marked as S0, S1, S2 but in part 2.3 these samples are referred on Fig. 5 and in Table 1 as P0, P1, P2. Please correct this error.

Lines 68 - 70 "TiO2 generates a photocatalytic effect that produces the reactive oxygen species (ROS) in degradation of various organic compounds." Probably responsible for  degredation... ?

Lines 73 - 74 "NPs exposed to UV radiation induces adverse effects" Should be "induce"

Lines 85 - 86. Words "The obtained materials" are repeated in two consecutive sentences. Please correct this part.

Please check the whole text carefully.

So this paper could be published after serious corrections and probably additional experiments.

Author Response

Dear Editor and Reviewers,

We would like to express our sincere gratitude to you and the reviewers for the effort, and precious time you spent reviewing our paper and providing valuable comments in your reports. It was your valuable and insightful comments that helped us to improve our review substantially. The authors have carefully considered the comments and tried their best to fulfil every one of them. We honestly hope that the manuscript meets your high standards after careful revisions. We have also included a point-by-point response to the reviewers and made the changes described above in the manuscript.

Reviewer 2

Dear authors,

  • the paper provides interesting results on antibacterial properties of the core-shell systems but there are some inconsistencies in interpretation of diffraction and XPS data. So, you have found chlorine containing phases (hydrophilite and hydrocalumite on Fig. 4) but chlorine on the surface of particles was not found by XPS or its spectrum was not presented in the paper.  So this point should be discussed in the new version of the paper. In particular, you shall check once more correctness of phases identification by XRD: every phase shall be identified basing on all characteristic lines not only one, these lines should be also marked even when they overlap lines of other phases.
  • Response: Thank you for your observations. The crystalline phases were initially identified by the XRD database. We correlated the results with literature data and the synthesis method. We resumed the identification of the XRD phases, marking the lines and when they overlap lines of other phases, respectively rutile which is the dominant phase. We also took into account the value of the phase percent, calculated by the diffractometer program. It was thus observed that the phase identified as hydrophilite (CaCl2) has only one more obvious line and its percentage is only 1%. Under these conditions, the probability of its presence is unlikely. We corrected the figure 4 in this regard. In the case of Hydrocalumite [Ca8Al4(OH)24 (CO3)Cl2(H2O)1.6(H2O)8], most of the lines were identified. Given the percentage of mass of chlorine in its molecule mass (6%) and the percentage of phase calculated by the program, the percentage of chlorine would be less than 1% and its dispersion could only be on the surface of commercial TiO2 particles which are usually stabilized with a protective layer containing compounds with Al and probable Cl. In the alkaline conditions of the surface treatment can results a compound identical or similar to hydrocalumite. In addition this surface of TiO2 is covered with the composite layer. Under these conditions it was not possible to identify chlorine through XPS.

  • It is necessary to provide the survey XPS spectrum as a figure in the paper before spectra of individual lines and upload this spectrum file and files of individual lines as a supplementary materials if possible in accordance to the good electron spectroscopy traditions.
  • Response: Thank you for your suggestions. We introduced the survey spectra and the files will be provided

  • In general, the whole section presenting XPS results should be significantly improved. Firstly I shall mention that selected spectrometer analyser mode with CAE = 50 eV cannot provide real high energy resolution: high resolution spectra could be recorded at CAE 5 to 10 but such mode requires much more spectra acquisition time to get good signal/noise ratio. From other hand, the selected resolution was adequate for this research, so please remove the words "high resolution" from the text.
  • Response: Thank you for your suggestions. We removed the words ‘high resolution’

  • On my mind, all figures from Appendix I should be moved to main text where they are discussed (if it does not exceed paper limits). All these figures should be presented in bigger size in the new version of the paper. Components of peaks corresponding to definite interatomic bonds should be labelled on spectra figures.
  • Response: We decided to remove the spectra for sample S0 and S1 since they do not make the object of this study in details for the XPS analyzea and they were not mentioned in the main text. We intend to present them in detail in a further study which will be the object of a new paper. Also, the spectra for Ti 2p was moved in the main text.

  • It is noticed in line 138 "two satellite peaks were observed at 438.53 eV and
    23 eV" so it is necessary to show them on the figure and discuss its origin from Cu2+in greater detail: according to NIST XPS database these lines could be Ca 2s.
  • Response: Thank you for your suggestions. Unfortunately, it was a mistake in the energies written in the text. The spectra for Cu 2p were modified and now it is presented the hole spectra in order to be clearer and the right energies are indicated. Also it was modified in the text: ‘Additionally, two satellite peaks were observed at 940.03 eV and 942.35 eV,…’

  • The paper contains too many inaccuracies, style errors and misprints. In the beginning of Section 2 "Results" the samples are marked as S0, S1, S2 but in part 2.3 these samples are referred on Fig. 5 and in Table 1 as P0, P1, P2. Please correct this error.
  • Lines 68 - 70 "TiO2 generates a photocatalytic effect that produces the reactive oxygen species (ROS) in degradation of various organic compounds." Probably responsible for.. ?
  • Lines 73 - 74 "NPs exposed to UV radiation inducesadverse effects" Should be "induce" –
  • Lines 85 - 86. Words "The obtained materials" are repeated in two consecutive sentences. Please correct this part. Please check the whole text carefully.
  • Response: Thank you for your observations. We corrected all the technical errors in the suggested lines.

Reviewer 3 Report

Photocatalysis is a promising technique with great potential for preparation of antimicrobial coatings. TiO2 is recognized as a traditional semiconductor photocatalyst, which can generate reactive oxygen species (ROS) under UV-light that cause degradation of various organic compounds. However, photocatalytic materials can be activated only under UV-radiation (which is toxic to humans) and it is of importance to modify their surface in order to extend the photocatalytic effect towards the visible region. Transition metals, introduced in the crystalline structure of TiO2 can help in this context.

The presented manuscript describes the synthesis and characterization of composite Ca and Cu coatings on TiO2 particles surfaces in order to extend the photocatalytic (resp. antimicrobial) activity of the system to the visible light. What is strange, however, the antimicrobial activities of the TiO2 particles coated with only Ca or with mixed Ca and Cu layers were found almost equal in the presence and in the absence of light (what happens with the photocatalytic effect of the TiO2 semiconductor?) what the authors considered to be a result from the electric field generated on the composite layers. Such interpretation is not discussed in detail (DFT part is presented quite schematic and not informative) and seems too unclear. Therefore, the conclusion that under visible light was evidenced the antimicrobial photocatalytic property and in dark the effect of electrical field generated by electrostatic potential of the composite layer needs better explanation before accepting this work for publication.

The manuscript also needs technical corrections:

l. 63 – Benzisothiazol is Benzothiazol;

l. 92 – Transfer electron microscopy is Transition;

Table 1 – P0, P1, P2 is S0, S1, S2;

1.154 and 157 – TiO (001) is TiO2(001);

1. 166 – Fig. 7 is the right number;

l. 445 – CuO2???

Ref. 20 is repeated;

Ref. 32 is not a Patent (see l. 392), etc.

Author Response

Dear Editor and Reviewers,

We would like to express our sincere gratitude to you and the reviewers for the effort, and precious time you spent reviewing our paper and providing valuable comments in your reports. It was your valuable and insightful comments that helped us to improve our review substantially. The authors have carefully considered the comments and tried their best to fulfil every one of them. We honestly hope that the manuscript meets your high standards after careful revisions. We have also included a point-by-point response to the reviewers and made the changes described above in the manuscript.

Reviewer 3

Photocatalysis is a promising technique with great potential for preparation of antimicrobial coatings. TiO2 is recognized as a traditional semiconductor photocatalyst, which can generate reactive oxygen species (ROS) under UV-light that cause degradation of various organic compounds. However, photocatalytic materials can be activated only under UV-radiation (which is toxic to humans) and it is of importance to modify their surface in order to extend the photocatalytic effect towards the visible region. Transition metals, introduced in the crystalline structure of TiO2 can help in this context.

The presented manuscript describes the synthesis and characterization of composite Ca and Cu coatings on TiO2 particles surfaces in order to extend the photocatalytic (resp. antimicrobial) activity of the system to the visible light.

Response: Thank you very much for your appreciated and valuable observation of our manuscript.

  • What is strange, however, the antimicrobial activities of the TiO2particles coated with only Ca or with mixed Ca and Cu layers were found almost equal in the presence and in the absence of light (what happens with the photocatalytic effect of the TiO2 semiconductor?) what the authors considered to be a result from the electric field generated on the composite layers. Such interpretation is not discussed in detail (DFT part is presented quite schematic and not informative) and seems too unclear.
  • Response: The information provided for the DFT calculation scheme and models of the three investigated systems is complete. The aim of the DFT calculations was to characterize the local electrostatic field in the region of TiO2(001) surface and to show the effects of the treatments of the rutile-TiO2(001) surface, by Ca and Cu2O film. We do not analyze other properties of the investigated systems in order to keep the manuscript simple and clear. The interaction mechanisms of virus/bacteria with the characterized electrical fields are not possible at the DFT level due to the huge required computational effort.

  • Therefore, the conclusion that under visible light was evidenced the antimicrobial photocatalytic property and in dark the effect of electrical field generated by electrostatic potential of the composite layerneeds better explanation before accepting this work for publication.
  • Response:Thank you for your constructive observation. It were introduced new explanation into Discussion section: “As can be seen in the UV-Vis spectrum, only the sample containing Ca and Cu shows absorption  in visible light. The antimicrobial photocatalytic effect of copper was highlighted for similar materials [17]. However, this study evidences a similar antibacterial effect in dark and light conditions for all the samples.  Given that only the sample with Ca and Cu can generate a photocatalytic effect and the results are similar (Fig. 11) we concluded that although it may be a competition between two processes, the one that determines the antimicrobial effect is that of the electric field (generated by electrostatic potential of the composite layer, see 2.4  section), present both in the dark and in the light.” 
  • The manuscript also needs technical corrections:
  • Response: Thank you for your observations. We corrected all the technical errors in the suggested lines.

Round 2

Reviewer 1 Report

Thanks for the answers.

Author Response

We would like to express our sincere gratitude to you for the effort, and precious time you spent reviewing our paper and providing valuable comments in your reports. 

Reviewer 2 Report

Dear authors,

the revised version of your paper was significantly improved but I still ask you to make some small corrections before publication:

1) Please change P2 to S2 in Fig. 5 caption (line 234)

2) Please label characteristic lines on the survey spectrum Fig. 5 (a): O 1s, Ti 2p1/2, Ti 2p3/2, etc. The "Survey spectrum" label should be corrected or removed.

3) Please label different chemical bonds on spectra of individual lines near corresponding BE values, for example, Cu (0), Cu+, Cu++  on Fig. 5 (c).

4) Please remove the words "high resolution" on lines 234 (Fig. 5 caption) and 553.

Author Response

We would like to express our sincere gratitude to you for the effort, and precious time you spent reviewing our paper and providing valuable comments in your reports. Please, find bellow, a point-by-point response to your observations.

Dear authors,

the revised version of your paper was significantly improved but I still ask you to make some small corrections before publication:

  • Please change P2 to S2 in Fig. 5 caption (line 234)

Response: Thank you for your observations. We changed the P2 with S2

  • Please label characteristic lines on the survey spectrum Fig. 5 (a): O 1s, Ti 2p1/2, Ti 2p3/2, etc. The "Survey spectrum" label should be corrected or removed.
  • Please label different chemical bonds on spectra of individual lines near corresponding BE values, for example, Cu (0), Cu+, Cu++  on Fig. 5 (c).

Response: Thank you for your observations. As the study compares the antimicrobial properties of the two samples (TiO2 microparticles coated with two different composite layers  with  Ca –sample S1 and with  Ca-Cu –sample S2)  and in order to follow your recommendations, to reduce the number of XPS figures and to abandon the figures from the Annex, we presented the XPS comparative results. We also consider that all the comments are better supported by the comparative presentation of the XPS results for the two samples. This is why we made the changes in Figure 5.

  • Please remove the words "high resolution" on lines 234 (Fig. 5 caption) and 553.

Response: Thank you for your suggestions. We removed the words ‘high resolution’
